# Establishment of the Bacterial Microbiota in a Lab-Reared Model Teleost Fish, the Medaka *Oryzias latipes*

**DOI:** 10.3390/microorganisms10112280

**Published:** 2022-11-17

**Authors:** Charlotte Duval, Benjamin Marie, Pierre Foucault, Sébastien Duperron

**Affiliations:** UMR7245 Molécules de Communication et Adaptation des Micro-Organismes, Muséum National d’Histoire Naturelle, CNRS, 75005 Paris, France

**Keywords:** microbiota, ecotoxicology, symbiosis acquisition, holobiont

## Abstract

*Oryzias latipes* is an important model organism for physiology, genetics, and developmental studies, and has also emerged as a relevant vertebrate model for aquatic ecotoxicology. Knowledge regarding its associated microbiota on the other hand is still scarce and limited to adults, despite the relevance of the associated microbiome to the host’s biology. This study provides the first insights into the establishment of bacterial microbiota during early developmental stages of laboratory-reared medaka using a 16S-rRNA-sequencing-based approach. Major shifts in community compositions are observed, from a Proteobacteria-dominated community in larvae and juveniles to a more phylum-diverse community towards adulthood, with no obvious difference between female and male specimens. Major bacterial taxa found in adults, including genera *Cetobacterium* and ZOR0006, establish progressively and are rare during early stages. Dominance shifts are comparable to those documented in another major model teleost, the zebrafish. Results from this study provide a basis for future work investigating the influence of medaka-associated bacteria during host development.

## 1. Introduction

In recent years, toxicology and ecotoxicology have embraced the so-called microbiome revolution [1,2]. A host-associated microbiota is now widely acknowledged not only as a functional, but also responsive compartment that must be accounted for in lab assays as well as environmental settings [3,4]. Indeed, microorganisms react to and interact with various contaminants and influence eventual adverse outcome pathways because they may sequester, inactivate, or a contrario amplify their effects for hosts. Various studies have demonstrated the effect of contaminants on bacterial community structure [5,6], as well as the capacity of members of the gut microbiota to metabolize various xenobiotics, with protective to enhanced toxic effects [7,8].

The medaka fish *Oryzias latipes* is a major nonmammal vertebrate model organism used for genetic, developmental, physiological, and ecotoxicological assays [9]. Thanks to its resistance, short life cycle, and amenability to lab maintenance, it has proven a valuable model to address the effect of various contaminants on freshwater fish, providing insights into the capacity of fish to cope with increasing anthropization and contamination of aquatic systems (e.g., [10,11,12]). Various tools have been developed on the medaka, including histological analysis, metagenome analysis, proteomics, transcriptomics and metabolomics. The effect of various contaminants on many aspects of fish physiology have been documented [10,13].

The medaka was already employed in several ecotoxicological experiments that investigated the response and influence of its associated microbiota [14,15]. A 28-day exposure to microcystin-LR-containing extracts from bloom-forming toxic cyanobacterial strains of *Microcystis aeruginosa* was shown to trigger major changes in gut-bacterial microbiota composition [16] and a dose-dependent response in microbiota variation, as well as significant effects on metabolites produced by the holobiont as a whole [17]. The gut microbiota is the fastest-responding compartment, with significant changes in relative abundances of taxa occurring within the first 24 h, emphasizing its potential importance for the holobiont’s response [18]. In these three studies, a limited number of intestinal bacterial taxa were shown to be major actors of the symbiotic system and its response, and they responded in a quite reproducible and predictable way. On one side, *Cetobacterium* was, for example, extremely stable whatever the changes; meanwhile, genus ZOR0006, the most abundant member of the Firmicutes, dropped in abundance upon exposure to cyanobacteria and their metabolites in all experiments.

These examples confirm the potential of lab-reared holobionts, such as the medaka, for ecotoxicological approaches that would include investigation of the microbiome’s response. However, establishing relevant model holobionts for microbiome-aware ecotoxicology requires a better understanding of host–symbiont interaction than currently available. Lab-reared animals are, for example, known to harbor only a limited fraction of the microbiota found in their wild counterparts, and that difference needs to be characterized [19,20]. Besides that, how a microbiota becomes established during the host’s life cycle needs to be addressed, because animals associate with microorganisms at all life stages [1,21]. The microbiome of early life stages in particular is identified as a potential target to mitigate diseases [22] and is a variable to consider in the (eco)toxicological assays that are conducted on eggs, larvae, and juveniles [23,24].

To support the relevance of the medaka fish as a model organism for microbiome-aware ecotoxicology, the aim of the present study is to understand how its bacterial microbiota is established throughout the different life stages in a laboratory setting. For this, bacterial partners are characterized using a 16S-rRNA-sequencing-based approach, starting from eggs to larvae, juveniles, up to sexually mature male and female specimens. Microbiota diversity and composition are explored in order to identify potential shifts in community structure as well as eventual differences between males and females. Altogether, this study provides new data regarding microbiota acquisition in an important model organism and will help account for microbial responses during early life stages.

## 2. Materials and Methods

### 2.1. Production of Life Stages

Adult specimens of medaka fish (*Oryzias latipes*) were maintained in freshwater-containing aquaria in the temperature-controlled medaka facility of the MNHN. Water parameters were monitored on a regular basis and adjusted (pH, conductivity, nitrates, and nitrites). Water was recirculated and replaced with freshwater (2/3 osmosis (RiOs 5, Merck Millipore, Burlington, MA, USA) and 1/3 filtered) by automated pumps. Fish were fed three times daily (~3–5% of the fish biomass per day) with Nutra HP 0.3 (Crude protein 57, Crude fat 17, N.F.E 7.5, Ash 10, Crude fiber 0.5, Phosphorus 1.7, Vitamins A, D3, E; Skretting, Stavanger, Norway). Reproduction was induced by shifting the light/dark cycle from 12:12 h to 15:9 h. Eggs were collected on day 0 postfertilization (dpf, d0, 9 batches of 3 eggs each). For later stages, other eggs were placed in a Petri dish containing Yamamoto medium (170 mM NaCl; 4 mM KCl; 3.6 mM CaCl_2_ 2H_2_O; 6.5 mM MgSO_4_ 7H_2_O; renewed daily), then placed in an incubator at 28 °C until hatching. Larvae were collected at hatching (d10 dpf, 9 specimens), and the remaining larvae were transferred to freshwater-containing aquaria with a light/dark cycle of 15 h:9 h for 1 week in a small tank, then in a larger one for the following 3 weeks.

Larvae were collected around the time of mouth opening when they start eating (d14 dpf corresponding to d4 post hatching, 5 specimens). After d30 posthatching (d40 dpf), fish were transferred to a new aquarium, with a 12:12 h light/dark cycle, and specimen sampling occurred at d30 (5 specimens). Later samplings allowed sex determination and occurred at d90 posthatching (4 males, 5 females) and d120 posthatching (7 males, 7 females). Reproduction was then induced by shifting the light/dark cycle from 12:12 h to 15:9 h after d90, confirming that the remaining fish had reached sexual maturity. Upon sampling, fish were anesthetized in 0.1% tricaine methanesulfonate (MS-222; Sigma, St. Louis, MO, USA) buffered with 0.1% NaHCO_3_ and sacrificed. Whole specimens were used from d0 to d30, while guts were dissected and analyzed for d90 and d120 specimens. Samples were flash-frozen in liquid nitrogen and stored at −80 °C. Also, 50 mL of water from aquaria was sampled at each date on a 0.22 µm nitrocellulose filter and frozen for DNA extraction.

### 2.2. DNA Extraction and Sequencing

DNA was extracted from water filters. For fish samples, batches of 3 pooled eggs forming the same clutch were extracted to maximize the amount of recovered DNA. DNA from whole individual specimens collected at d10, d14, and d30 were extracted due to small specimen size. Finally, DNA was extracted from dissected guts for d90 and d120 specimens. Extractions were performed using the ZymoBIOMICS DNA Mini-prep kit with a FastPrep 5G beat beater disruption (DNA Matrix; 4 × 30 s; 6 m.s^−1^). An extraction-blank control sample was also performed. The V3-V5 region of the 16S rRNA encoding gene was amplified using primers 341F and 926R [25] and sequenced on an Illumina MiSeq 2 × 300 bp platform (Eurofins Genomics, Konstanz, Germany). These longer reads encompassing the V3-V4 and V4-V5 regions were chosen because they cover a broad fraction of the prokaryotic diversity, with good resolution of clades such as Bacteroidia and Planctomycetes [26]. Reads were deposited into the Sequence Read Archive (SRA) database (BioProject PRJNA888957, samples SAMN31227644 to SAMN31227683; Appendix A).

### 2.3. Sequence Analysis

Merged raw reads were used (truncated to 523 bases) and quality-controlled, chimeras were removed, and Amplicon Sequence Variants (ASVs) were obtained with DADA2 (default parameters, length cutoff: 523 bp) using QIIME2 [27]. ASVs were affiliated using the SILVA 138-99 database, and ASVs assigned to Chloroplast, Mitochondria, Unassigned, and Eukaryota were discarded. Diversity metrics were computed with the phyloseq [28] R package (v1.40.0). Datasets were normalized by rarefaction to 2,013 reads, following recommendation [29]. Statistical analyses were performed using R packages vegan (v2.6-2) and PairwiseAdonis (v0.4) [30,31].

## 3. Results

Eggs hatched 10 days (d10) postfertilization, and fish larvae started feeding at d14 dpf. Juveniles sampled until d30 posthatching could not be assigned to male or female. Female and male specimens were distinguishable starting d90 posthatching, and fish were able to reproduce at d120.

### 3.1. Diversity of Bacterial Communities

Despite several attempts, no PCR product was obtained from any of the nine batches of three eggs, nor from any of the nine tested larvae collected at d10 dpf. Attempts to further pool eggs of larvae specimens did not yield PCR products either. Products were obtained from all specimens from d14 onwards, yielding a total of 650,196 paired-end quality-filtered reads (average 16,255 reads per sample, Appendix A) clustering into 722 ASVs. Rarefaction curves reached saturation for the selected rarefaction value (2013 reads), indicating appropriate depth of sequencing for identifying the dominant members of the bacterial community in further comparisons (not shown).

Species richness was highest in water (106.5 ± 28.4 ASVs, Appendix A, Figure 1). Among fish samples from the different sampling dates, species richness was significantly higher in d120 specimens compared to d14 and d90 (Kruskal–Wallis (KW) post hoc tests, *p* = 0.02 and *p* = 0.04, respectively, Figure 1, Appendix A), while the Shannon index was lower for d90 specimens compared to d30 and d120 (KW post hoc tests, *p* = 0.01 and *p* < 0.01, respectively, Appendix A). When considering sex, there was no significant difference either in species richness (KW test, *p* = 0.52, 12 females and 11 males, with d90 and d120 analyzed together) or in the Shannon index (KW test, *p* = 0.80).

### 3.2. Community Composition as a Function of Life Stage

Principal coordinates analyses using Unweighted (UU) and Weighted UniFrac (WU) distances clearly separated water samples from fish samples; among fish samples, the different sampling dates were clearly separated, with high explanatory power on the first two axes (38.8% in UU and 63.7% in WU, Figure 2). Comparison of bacterial community compositions using both UU and WU distances indicated significant differences among groups (PERMANOVA, *p* = 0.001 in both cases, Appendix A). Pairwise comparisons confirmed that the community composition in each group was significantly different from all others (PAIRWISE. PERMANOVA, *p* values <0.05, Appendix A). This indicates that both community membership as well as abundance differ according to life stage. On the other hand, compositions in intestines of males and females were not significantly different (PERMANOVA, *p* = 0.68 (UU) and *p* = 0.17 (WU), 12 females and 11 males sampled at d90 and d120 and analyzed together). Males and females clustered together in the graphs (Figure 2).

### 3.3. Community Structure and Dominant ASVs

Fish-associated bacterial communities were dominated by Proteobacteria at d14 and d30, with dominance of Gammaproteobacteria in the former, and relatively more abundant Alphaproteobacteria in the latter (Figure 3). A major shift towards dominance of Fusobacteria was then observed starting d90, with relative decrease in abundance of other groups, in particular Alphaproteobacteria. At d120, an increase was observed for Bacilli (phylum Firmicutes) and Actinobacteria.

At the ASV level, some taxa were present throughout all dates, including one gammaproteobacterial ASV affiliated to genus *Aeromonas*, found in most specimens at all dates (median 1.7 to 11.0% depending on life stage) and below 1% in the water samples (Appendix A). However, the abundance of most other major ASVs varied over time. Some ASVs tended to peak in abundance during one of the two earliest stages. Three ASVs, affiliated to the Comamonadaceae and to genera *Pseudomonas* and *Perlucidibaca*, were for example most abundant at d14 and then became much rarer (Appendix A). Other ASVs peaked at d30, including one assigned to genus *Legionella* (17.4 ± 14.7%), which was less abundant at d14 and far less abundant after d30. Two other ASVs, affiliated to *Hyphomicrobium* (Alphaproteobacteria) and to the Gemmataceae (Planctomycetes) also peaked at d30 and were absent before this date and rare after.

Major changes occurred at d90. The main ASVs responsible for the shift towards increased abundance of Fusobacteria were affiliated to genus *Cetobacterium*. This genus represented a median of 1.5 to 2.8% of reads in water communities at all dates. However, it was almost absent from animals before d90, but then became by far the most abundant taxon in most specimens (median 75.5 ± 12.6% at d90 and 52.6 ± 19.5% at d120). At d120, Fusobacteria (again mostly represented by genus *Cetobacterium*) and Proteobacteria were still abundant, but there was also a clear increase in Firmicutes, mostly consisting of 7 closely related ASVs belonging to genus ZOR0006. This genus was present in all specimens from d90 onwards (2.4 ± 1.5% at d90; 9.0 ± 6.0% at d120, Appendix A) but below 0.4% in specimens before this date and below 0.1% in water samples at all dates. In Actinobacteria, one ASV belonging to genus *Nocardia* also appeared at d90 in low numbers (0.5 ± 0.4%) and considerably increased at d120 (5.3 ± 10.0%), as observed in water samples, while it was absent from earlier stages. Besides these main ASVs, other ASVs were detected in most specimens starting at d90 and were then stable, including one affiliated to *Shewanella* and several others that were present in low abundances. No abundant ASV was sex-specific.

Despite these changes in abundances, many of the identified ASVs were shared between different life stages (Figure 4). When considering only abundant ASVs (i.e., ASVs that represented at least 1% of reads in at least one sample), 68 were shared between all life stages.

## 4. Discussion

The medaka fish *Oryzias latipes* is a major vertebrate model organism for ecotoxicology as well as molecular biology, and addressing the composition of its associated microbiota at all life stages is necessary in the context of the emerging holobiont approach [19,32]. Indeed, descriptions are already available for other major aquatic models, including the zebrafish [21,33,34]. In this study, specimens from the MNHN medaka facility were employed. The facility was established in 2005, and the parents of all specimens used were obtained from a single batch originating from a reference facility in France.

### 4.1. Bacteria Become Detectable Only after Mouth Opening

Despite repeated attempts and pooling of several specimens, no PCR product could be obtained from eggs and d10 dpf larval stage (at hatching), suggesting either the absence of bacteria or their occurrence at levels below the detection limit of the chosen PCR strategy. These stages correspond to developmental steps that predate mouth opening and the start of external food uptake. Thus, at these stages, colonization of the digestive tract through oral uptake cannot have started yet. It was previously shown that a very low abundance of bacteria occurred on the skin and gill of adult lab-reared medaka fish, based on the need to pool several samples to obtain a positive PCR response and on fluorescence in situ hybridization assays (S. Duperron, pers. obs). This lack of efficient colonization of external epithelia could be linked to the relatively clean rearing conditions, which involve regular water renewal by a mixture of osmosed and filtered water, certainly leading to low bacterial abundance in the water. This assumption is also supported by the very low DNA yields obtained from water filters and from d0 and d10 samples here. Regardless, this certainly is specific to our lab-reared conditions. Indeed, colonization of the egg and skin surface is well-documented in aquaculture settings; for example, for commercial species such as the European Seabass and Gilthead Seabream, and in environmental settings as documented from the brown trout and some coral reef fish [35,36,37]. In a study of zebrafish from a lab rearing facility, positive PCR signals were obtained as early as 4 dpf [21]. In our study, bacteria become detectable for each individual from d14 dpf onwards, indicating that densities reach a minimal level shortly after opening of the mouth.

### 4.2. Community Shifts Occur throughout Development

Bacterial diversity levels tend to increase only marginally at the latest (d120) compared to earlier stages, but remain within the same order of magnitude, and evenness does not vary much, indicating that no huge variation in the overall level of fish microbiome diversity occurs during development. The first two stages in which bacteria are detectable, namely d14 and d30, were analyzed as whole specimens, their gut being too small to be dissected and analyzed. In these, Proteobacteria are dominant, mostly Gamma- then Gamma- and Alphaproteobacteria. Temporal shifts in dominant ASVs are also observed, mostly involving ASVs related to waterborne bacteria, many of which belong to groups previously documented in fish eggs (Comamonadaceae, *Pseudomonas*), and some of which may also include potential opportunistic pathogens (e.g., *Aeromonas*, *Pseudomonas*, *Legionella*, *Nocardia*, *Shewanella*) [12,35,36,38]. In specimens from d90 and d120, only the gut microbiome was analyzed. Various works have shown that the fish-gut-associated microbiota is very different from that of surrounding water, and that skin- and gill-associated communities may display intermediate composition [39]. Change in the target organ could thus explain some of the observed shifts in community composition, because the gut can be considered a much more selective habitat compared to external surface tissues [40]. However, similar dominance of Gammaproteobacteria until 10 days postfertilization (dpf) followed by an increase in the abundance of Alphaproteobacteria at 28 dpf was described in dissected intestine of lab-reared zebrafish, suggesting that our observations are not only explained by the use of whole specimens instead of dissected gut in the earliest stages [21]. Moreover, dominance of Proteobacteria is documented in eggs and early life stages of various commercial fish species [37] as well as wild species, including the brown trout [35], Atlantic salmon [22], and two coral reef fish [36], indicating that this could be a general trend in teleosts.

In the present study, from d90 onwards, Fusobacteria of the genus *Cetobacterium* become dominant, and Firmicutes also become abundant, mostly consisting of genus ZOR0006. Major changes in community composition were also documented between 35 and 75 dpf in the zebrafish, with dominance of Fusobacteria and Firmicutes from 75 dpf onwards [21]. *Cetobacterium* and ZOR0006 are two of the most dominant gut-associated bacteria identified in all our previous studies on the medaka from the MNHN facility [16,17,18]. *Cetobacterium* is a major nonpathogenic gut-resident bacterium in many fish species, including zebrafish and tilapia [33,41,42,43]. It is reported to be highly abundant in the gut of wild medaka [12]. It was shown to drop in relative abundance upon exposure to the antibiotics ampicillin and erythromycin but to be resistant to exposure to cyanobacterial stress in the lab [12,17]. Its genome encodes various functions relevant to the host, including heterotrophic metabolism and vitamin B12 production [17]. Similarly, ZOR0006 is a gut specialist bacterium of which the abundances were shown to drop dramatically upon exposure to the same cyanobacterial metabolites and with abilities to degrade various carbohydrates [17]. A drop in the relative abundance of either of these two dominant bacteria is considered a sign of dysbiosis in medaka fish [12,18]. Interestingly, an unclassified Firmicutes (CK-1C4-19) was also shown to become abundant in adult zebrafish while unabundant in earlier stages, suggesting similarities between the two model species [21]. Besides these two dominant genera, a diversity of not necessarily abundant ASVs is also observed only at these later stages, suggesting a community enriched in additional gut specialists. Overall, medaka sampled at d120 display a gut-associated community highly similar to that previously documented in older adult specimens reared in the same facility, suggesting that the community converges towards a certain level of stability by then [18]. Interestingly, community shifts observed herein are very comparable in terms of taxa involved and succession to those documented in the zebrafish [21,33], and they also display some congruency with other investigated nonmodel fish species, which could suggest general trends in teleost fish.

### 4.3. No Evidence for Differences between Males and Females

Specimens from d90 and d120 could be assigned to pubertal or mature male or female. Their associated communities were not significantly different at a given date though, either in terms of diversity or in composition. Sex-dependent effects are documented in microbiota composition of vertebrates, assumed to be due to hormones, reproductive status, and immune response [44,45]. On the other hand, no difference was identified between gut microbiomes of males and females in the zebrafish at 75 and 380 dpf [21]. Our results are thus congruent with those obtained on this major model teleost. However, for future studies, one should keep in mind that sex-specific effects in microbiota responses are still possible. They have been documented in the microbiota of zebrafish exposed to nanoparticles, for which only males displayed composition change [46].

## 5. Conclusions

Bacterial communities associated to lab-reared medaka were not detectable until after mouth opening, possibly due to the clean environment. After that, community compositions changed with time, with shifts in community compositions that resemble those documented in lab-reared zebrafish, another major model teleost. They converge towards the composition found in guts of adults upon sexual maturity, with no significant differences between males and females at least until day 120. Future studies investigating effects on the medaka holobiont during its development, for ecotoxicology as well as other research fields, must thus consider the changes that occur in microbiota composition, which is largely a function of life stage. This will be particularly important in tests that involve the earliest stages.

## Figures and Tables

**Figure 1 microorganisms-10-02280-f001:**
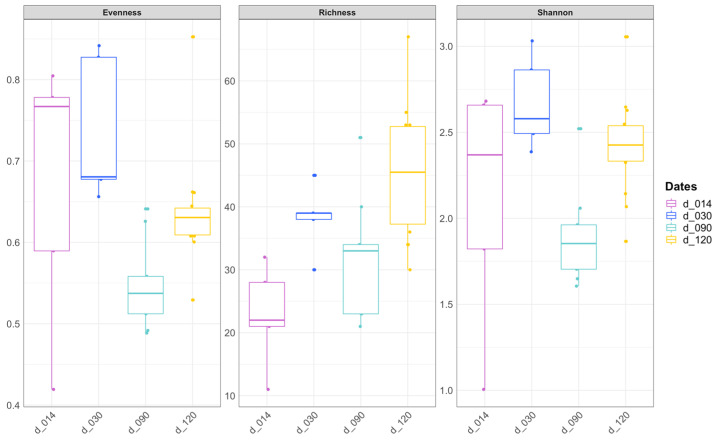
Evenness, observed species richness and Shannon index for fish specimens collected at the different dates (d14, d30, d90 et d120).

**Figure 2 microorganisms-10-02280-f002:**
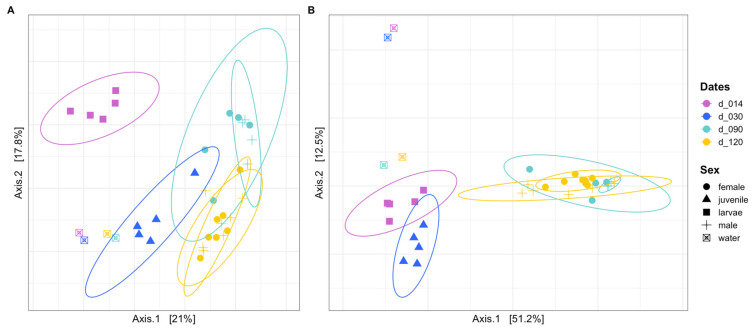
Principal coordinates analyses based on the unweighted (**A**) and weighted (**B**) UniFrac distances, illustrating the bacterial compositions from water and fish samples according to sampling date.

**Figure 3 microorganisms-10-02280-f003:**
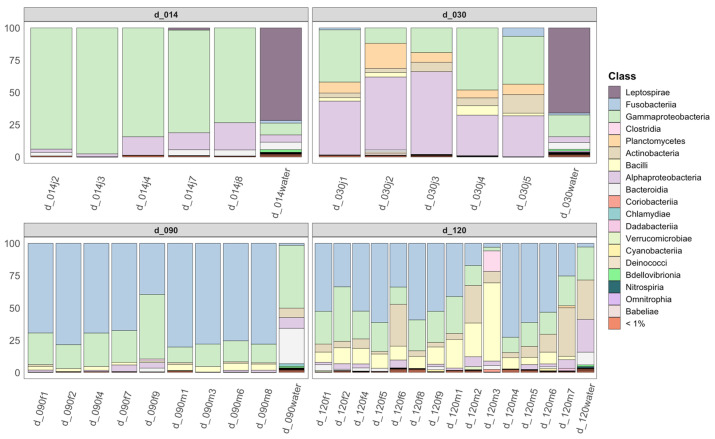
Relative abundance of the bacterial taxa at the class level across individual fish and water samples from the different dates. All taxa with abundance <1% in the whole dataset are grouped. Nomenclature according to Appendix A.

**Figure 4 microorganisms-10-02280-f004:**
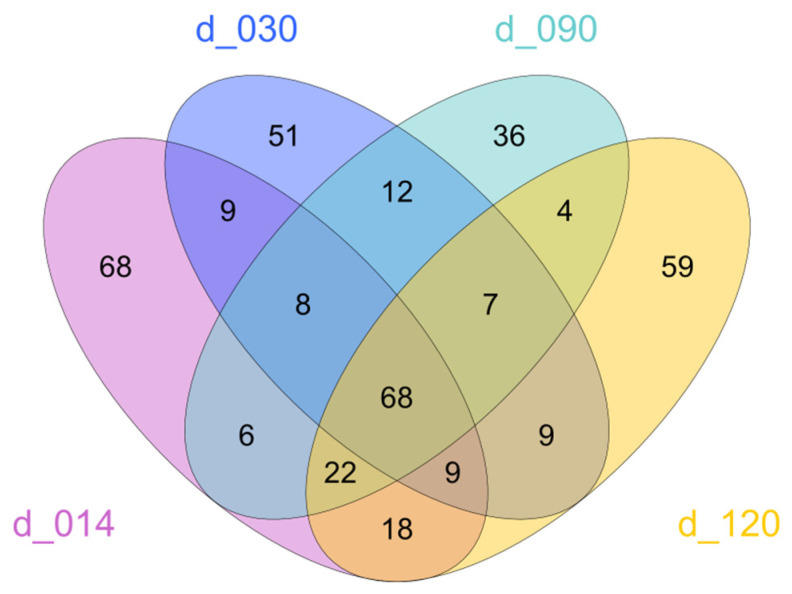
Venn diagrams displaying Amplicon Sequence Variants (ASVs) obtained from fish samples at d14, d30, d90, and d120, including only abundant ASVs, i.e., ASVs that represent at least 1% of filtered reads in at least one sample.

## Data Availability

Raw Illumina reads were deposited into the Sequence Read Archive (SRA) database (BioProject PRJNA888957, samples SAMN31227644 to SAMN31227683; Appendix A).

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
