# Peer review of "Establishment of the Bacterial Microbiota in a Lab-Reared Model Teleost Fish, the Medaka *Oryzias latipes"

_microorganisms, 2022, doi:10.3390/microorganisms10112280_

Round 1

Reviewer 1 Report

The authors investigate the gut microbiome changes across different host ages, and found the potential effects on the microbial community by the ages. I have some concerns on this study.

(1) The authors mentioned the importance on understanding the gut microbial community in this fish. However, the authors only used 16s v4 region, which method only be confident on the genus level. The 16s full length sequencing will provide the species, and even the strain information for this fish. 

(2) There are no functional analysis on this study. the gut microbial composition and function are the two basic information on the animal gut microbiome. Thus, if possible, the metagenomes of this fish will help readers understand this lab-reared animal model. 

Author Response

Dear editor,

We thank reviewers for their helpful reports, and you for handling our manuscript. Please find below our answers to their comments, along with a version of our manuscript with changes marked for the sake of clarity, and a revised version.

Besides requests from reviewers, we also have modified a few sentences and added some additional detail where appropriate to clarify the manuscript. This includes presenting median percentages of specific important genera that we present in the results and that were not provided previously (for this, we added table S4). We also indicated the types of statistical tests (see version with changes marked). We also have improved the quality of figures for better rendition on the manuscript and homogeneisation of labels.

We hope this revised version will be deemed suitable for publication in Microorganisms, and are looking forward to your decision.

On behalf of authors,

Pr. Sébastien Duperron

Reviewer 1

The authors investigate the gut microbiome changes across different host ages, and found the potential effects on the microbial community by the ages. I have some concerns on this study.

  • The authors mentioned the importance on understanding the gut microbial community in this fish. However, the authors only used 16s v4 region, which method only be confident on the genus level. The 16s full length sequencing will provide the species, and even the strain information for this fish. 

We agree with reviewer that full length 16S rRNA encoding gene is a better predictor of taxonomic affiliation of bacteria versus only variable region. Indeed, shorter reads are less reliable than longer reads. It is not yet possible to get high number of full-length reads for moderate prices, and according to common practice in 16S studies, exemplified by recommendations from the Earth Microbiome Project (https://earthmicrobiome.org/) we relied on the sequencing of hypervariable regions as done in most studies to date.

However, as reviewer accurately points out, longer reads are a better choice. This is the reason we chose to perform Illumina MiSeq sequencing on a 2*300 bp (see l. 110) and using the V3-V5 region. Recently, papers have compared the V3-V4 (one of the most extensively used region) and V4-V5 regions and have shown that they overall perform similarly, with a good description of the community, but with different resolution of different groups. The class Bacteroidia is better described by V3–V4, and the phylum Planctomycetes by V4–V5, both clades which occur in our fish samples. The choice of the whol longer V3-V5 region thus seems the best possible for our study. This has been indicated l. 100 (version with changes marked) : “These longer reads covering the V3-V4 and V4-V5 regions were chosen because of their ability to cover a broad fraction of the prokaryotic diversity with good resolution of clades such as Bacteroidia and Planctomycetes [26]. » and an additional paper was cited (Fadeev et al, 2021).

  • There are no functional analysis on this study. the gut microbial composition and function are the two basic information on the animal gut microbiome. Thus, if possible, the metagenomes of this fish will help readers understand this lab-reared animal model. 

Unfortunately, it was not possible to obtain genomic data from individuals due to the high host-to-microbe DNA ratio, and a functional analysis was out of the scope of this study. However, we have some preliminary data from adults from a different study that support some published assumptions about close relatives of identified dominants ASVS. For example, the production of B12 vitamin by genus Cetobacterium. As suggested by reviewer, we extended a bit the discussion to address possible functions and roles of the gut microbiota. The following sentences were added:

L254: “Aeromonas, Pseudomonas, Legionella”

  1. 274 “Cetobacterium is a major non-pathogenic gut resident bacterium in many fish species including zebrafish and tilapia [32, 40–42]. »
  2. 276 « It was shown to drop in relative abundance upon exposure to antibiotics ampicillin and erythromycin” to provide a more detailed description of the results from previous studies”
  3. 282 the link with dysbiosis is now explicited: “A drop in the relative abundance of either of these two dominant bacteria is considered a sign of dysbiosis in medaka fish [12, 18]. »
  4. 291 “Interestingly, community shifts observed herein are very comparable in terms of taxa involved and succession to those documented in the zebrafish [21, 32], and display some congruency with other investigated non-model fish species that could suggest general trends in teleosts »

Reviewer 2 Report

The manuscript entitled “Establisment of the bacterial microbionta in a lab-rearedd model teleost fish, the medaka Oryzias latipes" provides information about the microbiota established throughout the different life stages of medaka in a laboratory setting. The work presents the originality of the results and the relevance of the subject The manuscript is well written and structured. The methodology is adequate and explicitly stated. The quality and quantity of presented data are completed. The discussion section is completed and very well argued.

This is a very interiesting paper.

Author Response

Reviewer 2

The manuscript entitled “Establisment of the bacterial microbiota in a lab-reared model teleost fish, the medaka Oryzias latipes" provides information about the microbiota established throughout the different life stages of medaka in a laboratory setting. The work presents the originality of the results and the relevance of the subject The manuscript is well written and structured. The methodology is adequate and explicitly stated. The quality and quantity of presented data are completed. The discussion section is completed and very well argued.

This is a very interiesting paper.

We thank reviewer for their positive comment; we carefully edited the text to check for typos and things that could be improved (see version with changes marked and reply to reviewer 1 and editor).

Round 2

Reviewer 1 Report

The authors have addressed most of my comments. The functional data will be informative in future studies in these fishes.